# Living with Rheumatic Heart Disease at the Intersection of Biomedical and Aboriginal Worldviews

**DOI:** 10.3390/ijerph19084650

**Published:** 2022-04-12

**Authors:** Emma Haynes, Minitja Marawili, Makungun B. Marika, Alice Mitchell, Roz Walker, Judith M. Katzenellenbogen, Dawn Bessarab

**Affiliations:** 1School of Global and Population Health, University of Western Australia, Crawley, WA 6009, Australia; roz.walker@uwa.edu.au (R.W.); judith.katzenellenbogen@uwa.edu.au (J.M.K.); 2Centre for Aboriginal Medical and Dental Health, University of Western Australia, Crawley, WA 6009, Australia; dawn.bessarab@uwa.edu.au; 3Menzies School of Health Research, Casuarina, NT 0810, Australia; minitja.marawili00@gmail.com (M.M.); makungun.marika@gmail.com (M.B.M.); alicem404@gmail.com (A.M.); 4Ngangk Yira Institute for Change, Murdoch University, Murdoch, WA 6150, Australia

**Keywords:** First Nations Australians, Aboriginal Australians, Indigenous Australians, Aboriginal ways of knowing, being and doing, rheumatic heart disease, biomedical worldview, colonisation, wellbeing, empathy

## Abstract

Rheumatic heart disease (RHD) significantly impacts the lives of First Nations Australians. Failure to eliminate RHD is in part attributed to healthcare strategies that fail to understand the lived experience of RHD. To rectify this, a PhD study was undertaken in the Northern Territory (NT) of Australia, combining Aboriginal ways of knowing, being and doing with interviews (24 participants from clinical and community settings) and participant observation to privilege Aboriginal voices, including the interpretations and experiences of Aboriginal co-researchers (described in the adjunct article). During analysis, Aboriginal co-researchers identified three interwoven themes: maintaining good feelings; creating clear understanding (from good information); and choosing a good djalkiri (path). These affirm a worldview that prioritises relationships, positive emotions and the wellbeing of family/community. The findings demonstrate the inter-connectedness of knowledge, choice and behaviour that become increasingly complex in stressful and traumatic health, socioeconomic, political, historical and cultural contexts. Not previously heard in the RHD domain, the findings reveal fundamental differences between Aboriginal and biomedical worldviews contributing to the failure of current approaches to communicating health messages. Mitigating this, Aboriginal co-researchers provided targeted recommendations for culturally responsive health encounters, including: communicating to create positive emotions; building trust; and providing family and community data and health messages (rather than individualistic).

## 1. Introduction

Rheumatic heart disease (RHD) significantly impacts the lives of First Nations Australians (see [1] for discussion of terminology), and is a potent marker of health inequity. RHD is chronic damage to the heart valves resulting from acute rheumatic fever (ARF), an abnormal autoimmune reaction to Group A Streptococcal (‘Strep A’) infection of the throat or skin [2]. The term RHD is used in this article to refer broadly to all the disease stages (see Table 1) unless otherwise specified. RHD occurs at exceedingly high rates in Indigenous communities worldwide. Despite its severity and preventability, RHD receives insufficient global attention and resources [3]. Approximately 420 cases of ARF are diagnosed among First Nations Australians annually (RHD Australia, 2012), a rate 123 times greater than in non-Indigenous Australians [4]. Further, of the more than 5000 people under 55 years of age living with RHD in Australia, 71% are First Nations Australians [4]. This represents one of the highest burdens of RHD per population in the world. The consequences of ARF and RHD are far reaching, significantly impacting, as well as shortening, the lives of First Nations children, adolescents, young adults, individuals, families and communities in Australia [5,6,7]. Table 1 summarises the key stages in RHD progression and prevention/treatment strategies.

While the commitment and advocacy of health practitioners and researchers have resulted in notable improvements, systemic and structural impediments constrain widespread gains. This lack of progress is likely to continue while biomedical and epidemiological understandings are privileged, and relevant First Nations knowledge, lived experiences and opinions remain poorly understood. Arguably, part of the problem is a lack of attention to the voices and understandings of the people behind the statistics. The use of statistical comparisons is ‘unlikely to deviate from well-worn themes of disadvantage and deviation from the norm’ [9], with an ‘inherent potential to underpin pejorative discourses of Aboriginal lived reality’ [9]. This leads to problematising the health of First Nations people without considering either their strengths or how the impacts of colonisation and the dominance of biomedical discourses continue to influence their ongoing health narrative [10]. Importantly, this includes acknowledging the legitimacy of First Nations knowledge and practices for healing.

To address this understanding, the lived experience of children, adults and communities with RHD was prioritised by the RHD clinical and research community in the NHMRC-funded End RHD Centre for Research Excellence (CRE) (GNT1080401). The CRE part-funded a PhD study ‘lived experience’ component, as formulated by Aboriginal Chief Investigators and Associate Investigators to generate greater understanding and improve outcomes [2]. We report here on this research. As RHD ‘predominantly affects the poor and underprivileged in society, the very people whose voices are unlikely to be heard in the absence of strong advocates of their cause’ ([2], p. 24), our research aims to ensure that lived experience stories become meaningful tools for advocacy and action—that is, to broaden the space of productive dialogue so that the voices of people living with RHD are heard and provide an alternative to solely biomedical narratives [1]. 

## 2. Materials and Methods

The PhD study reported here was undertaken in a partnership with Yolŋu community members and informed by the qualitative, decolonising approaches reported in our companion paper in this issue [1]. Applying this approach to the study design ensured that all data collection methods incorporated culturally based practices, including yarns [11,12], *nhina, nhäma ga ŋäma* (sit, listen, observe) [1] and informal conversations, and the analysis privileged Aboriginal voices. These methods are examined in detail in the companion article [1]. This meant not only hearing the perceptions, knowledge and understandings of children, teenagers, young adults, adults, families and communities living with ARF/RHD but also including the interpretations and experiences of Aboriginal co-researchers to inform the data analysis and reporting. Within this methodological stance, the study design also reflects a conventional, focused ethnographic approach. 

### 2.1. Recruitment

Recruitment for yarns took place in two clinical settings and two community settings in 2017 in the Northern Territory (NT) of Australia, resulting in 24 yarns. Purposive and snowball sampling were used to gain representativeness. All participants consented to be interviewed after having the research explained verbally, in addition to being given a patient information sheet. At two sites, the invitation to participate and an explanation of the project was given in Yolŋu *matha* (language) by Yolŋu co-researchers. Yarns were conducted in a variety of settings—clinical environments, beaches, under trees, at cafes and in homes. Many yarns involved parents and other family members—aunts, siblings, grandmothers—were often invited to join us because they were nearby and interested. 

### 2.2. Data Collection 

The yarns lasted between 15 to 90 min and were predominantly in English except for some sites, where a Yolŋu co-researcher was present. In those instances where the interviews moved between Yolŋu *matha* and English, the former were either translated by the fourth author (AM) (applied linguist) or by Yolŋu co-researchers and then transcribed. When participants declined to have the yarn recorded, notes were immediately typed up.

Additional data collection was guided by the practice of *nhina, nhäma ga ŋäma* [1]. Operating in a culturally appropriate manner, asking few questions and instead reflecting on the possible feelings embodied in the actions and conversations being observed, which resulted in nearly 200,000 words of electronic notes and three hard-copy journals. These observations, in turn, led to reflective conversations with Yolŋu co-researchers, particularly regarding feelings and the use of metaphors. Relevant journal notes were coded and, after having established themes and sub-themes, were included where they added value to the findings. The Yolŋu co-researchers also contributed understandings related to the lived experience of RHD based on their own *nhina, nhäma ga ŋäma* (participant observation).

### 2.3. Analysis 

The thematic analysis was an iterative process, commencing with fieldwork, involving the first author (EH) and three groups of Yolŋu co-researchers. The fourth author (AM), a trained linguist, Yolŋu *matha* speaker and qualitative researcher, helped facilitate the analysis process with the Yolŋu co-researchers. This meant that the Yolŋu co-researchers’ thinking and contribution to the analysis processes were confirmed in Yolŋu *matha*. Throughout the analysis process, Yolŋu and *balanda* (non-Aboriginal) co-researchers made a space for productive dialogue by applying both-way learning and *nhina, nhäma ga ŋäma*, both in communications with each other and in the reading, listening to and interpreting of the yarns. 

While every effort was made to give equal value to all participant stories, some voices came through more strongly than others. There are several important reasons for this. Some participants spoke as both a parent/carer and as a patient (differentiated at the end of quotes in the results section). Further, several yarns were significantly more in-depth due to familiarity because of the interviewer’s community immersion and/or to the participant’s confidence speaking in English, allowing a greater sense of productive dialogue. While a community translator was used for many yarns in one community, this did not always help in resolving communication difficulties.

### 2.4. Ethics

Ethics approval was obtained from the Human Research Ethics Committee (HREC) of the Northern Territory (NT) Department of Health and Menzies School of Health Research. Approval number HREC 2016–2678. Reciprocal approval by UWA HREC Office, Ref: ROAP 2020/ET000283. The following approvals were obtained prior to commencing the research: Ochre card (NT Working with Children); Northern Land Council (permit to reside in a homeland); written approval from the Aboriginal elders of the community and the local Health Service.

## 3. Results

### 3.1. Participant Characteristics

All 24 participants were Aboriginal, the vast majority being Yolŋu. The nature of participants’ ARF/RHD experiences covers a broad range of medical conditions, from uncomplicated ARF cases to complex valvular disease, including heart failure. Seven participants (four female, three male) were carers of children/adolescents with ARF/RHD. Two of these also reflected on their own ARF experiences. Nine of these were male participants, and 15 were female. This sample is a good reflection of the prevalence of RHD in Australia, where females form two-thirds of cases [4]. Sadly, two participants are now deceased. Pseudonyms are used to maintain anonymity (See Appendix A for the table of participants).

### 3.2. Themes

The Yolŋu co-researchers described the thematic analysis as making a ‘mat for everyone to sit on’; that is, the findings need to be applied and useful. The themes identified by Yolŋu co-researchers affirm an Aboriginal worldview of interconnection between mind, spirit and body and its relationship with praxis and good decisions as it relates to health and wellbeing [13]. The theme names identified by Yolŋu co-researchers have a deceptive simplicity, in the use of English as a third or fourth language. In fact, the Yolŋu are adept at using conceptual language, where simple terms hold a depth of knowledge that is only slowly revealed as the hearer is considered ready for greater complexity or depth [1]. The themes explored in detail below are summarised below (Figure 1).

In the early stages of data analysis, a theory to describe the interactions between these three themes was described graphically by Yolŋu co-researchers and the author (MM). Reflecting their holistic worldview, the Yolŋu co-researchers saw the themes as interwoven, spiralling feedback loops, as good feelings based on clear understanding led to good choices (or conversely, bad feelings, lack of understanding and poor choices). The decisions theme is the nexus, the point at which what one understands and how one feels (mind and spirit) intersect to determine actions (Figure 2, below).

This diagram became the lens that allowed us to see the significant data in the yarns. As new stories were added, it became increasingly evident how the relatively simple relationship described in the graph becomes very complex given interpersonal, family and community relationships along with socioeconomic, political, historical and cultural contexts, particularly with the overlay of the biomedical approach of practitioners. As a consequence, individual stories cannot be allocated to a static position on the diagram; rather positions are fluid depending on a range of influences. Each of the three themes is discussed in more detail below.

### 3.3. Maintaining Good Feelings

The findings related to maintaining good feelings reveal a distinctive Yolŋu worldview of connectedness to family, community and country through feelings as one aspect of cultural, social and emotional wellbeing. Maintaining good feelings was the first theme identified by the Yolŋu co-researchers. This prioritisation is no surprise as they often talked about the ‘importance of how you’re feeling, purpose, and identity… and being able to talk about feelings’ (Journal note, 31 October 2016). Being attuned to how others are feeling (having empathy) is a central feature of Yolŋu life. Feelings can be both an emotional state and a physical feeling. Three sub-themes are explored.

#### 3.3.1. Creating Good Feelings 

All Yolŋu learning stems from understanding one’s *gurrutu* (kin relationships) [14], beginning with learning the importance and obligation to take care of others’ feelings. The importance of maintaining good feelings (which, as indicated later, may involve some denial or a refusal to talk about RHD) is central to the Yolŋu worldview, influencing behaviour and actions. This sub-theme includes examples of the support given and received from both having regard for the feelings of others and by having one’s feelings considered. 

##### Good Feeling Support from Family 

Support from family is a principal source of good feelings. Demonstrating the importance of family support, teenage girl Dhumdhum came to her interview with her mother, sister, baby brother and aunty. Dhumdhum’s aunty described that connection and empathy with family was healing ‘like medicine.’ Similarly, Rinytjan, the father of a young man with ARF, described that ‘healing takes place is, from us, from Yolŋu… from person to a person… we have to be part of it… I think it’s more physical or you can do it by talking’ (Rinytjan, father). Importantly, for Yolŋu, just being present in a time of need is supportive, a therapeutic form of *nhina, nhäma ga ŋäma*, whereas Donatis (2010) states ‘the presence of the ancestors and of other spirit-beings is strongly felt and becomes part of the reality directly experienced by those engaged in the situation, even when their participation consists, apparently, by simply “being there”’ ([15], p. 97) [1]. Thus, being present is a requirement, a responsibility that comes with family relationships.

As will be discussed further in the final theme, the influence of family on an individual’s decision-making process is significant, often prioritised over other considerations. 

##### Good Support from Other Sources

Good support can come from sources other than family and community, and can be practical through the sharing of knowledge, experience or skills. For example, support from school and health services. The culturally appropriate and friendly support from Aboriginal health workers contributes to people with RHD feeling good and confident in the care they receive. Conversely, as Guya, the mother of a teenage participant, described health workers who leave can set people ‘back to square one’.

##### Sharing Helps Maintain Good Feelings

Taking care of others’ feelings is linked to sharing obligations that help connect people (for example, food and tobacco). Significantly, this relates to the observation by one of the local clinic nurses that it was better to give gifts to build relationships before asking teenagers to come for an injection rather than the more common use as an incentive or reward after the injection. Participants also spoke of being able to support others through sharing their experiences.

#### 3.3.2. Avoid Feeling Bad 

As part of maintaining good feelings, it is important to not allow yourself to feel bad and to protect others from feeling bad. Feeling bad is seen to be communicable; one negative person can cause everyone in the community to feel bad. A Yolŋu co-researcher explained that in order to ‘stay feeling good, keep away from people who make you feel bad—don’t accept it’, for example, ‘a doctor can make you feel bad’ (with a negative story about your health). The significance of bad feelings as causing, or perhaps exacerbating, poor health was evident in many yarns.

Discussing how *yätj ŋayaŋu* (bad feelings) can cause sickness, a Yolŋu co-researcher reported that when they first heard health messages in Yolŋu *matha* on the radio, people asked, ‘Are they trying to kill us?’. ‘This bad feeling is more than just emotional; it affects us physically and makes us lose confidence’ [16]. Given that these radio programs involved careful translation and good intent, the negative response from the Yolŋu highlights the complexity of creating a space for productive dialogue between Yolŋu and *Balanda* health practitioners. 

Strategies to avoid or manage situations that might cause bad feelings and ways of acting to protect others from feeling bad are discussed below. 

##### Not Letting Yourself Feel Bad 

Avoiding feeling bad (which may include feelings of fear, pain or grief) takes precedence over other, less immediate outcomes. For example, despite previously expressing to clinicians a willingness to have regular injections, avoiding feeling bad may override this commitment. Participants spoke of managing bad feelings; for example, when Rinytjan, who had RF as a teenager, stated, ‘I’m not going to feel sick… not believe I had this condition’. This can extend to refusing to answer questions, such as when Dankapa was completely silent in response to being asked how she felt about deciding whether or not to have an operation, and her daughter said, ‘It makes Dankapa feel bad to [be] talking about it.’ This reluctance is underpinned by the trauma associated with knowing that having RHD had ended badly for other family or community members. While this might appear to an outsider or health practitioner to be an irrational denial of ‘the facts’, from a Yolŋu worldview, the preservation of good feelings is an understandable response to avoid feeling bad. 

This reluctance to talk about being sick sometimes made it difficult to ask questions about health in the yarning sessions. The reluctance to talk about one’s poor health is similar to the refusal to talk about *galka* (sorcery), for fear that talking about it will make it happen. 

##### Healthcare as a Source of Feeling Bad

Clinicians giving secondary prophylaxis injections can easily cause pain and/or lasting trauma due to lack of training, other clinic demands or not making a pain-minimised injection a priority. As evident in the interview findings, these traumatic experiences stay with many patients and impact their decision-making around treatment; as Guya spoke of her own ARF experiences ‘They had to hold me down… I hated it… I put up a fight, I think I just had one too many bad experiences’. Lacking an understanding of how enduring the memory of these painful experiences can be, clinicians are likely to compound the trauma by poorly conveying information about the necessity for injections in a way that is likely to make the patient scared and worried. 

Further, as clinicians are sometimes involved in child removals, some families are fearful of having their children taken away when seeking healthcare. This remains a very present fear (evidenced by refusals to participate in interviews in one of the clinics), which reflects a transgenerational trauma [17]. Similarly, going to hospital is a particular source of feeling bad. While potentially resulting in actions such as refusals and silences, a strengths-based lens in this context recognises that choosing to feel good is an assertive act, particularly when one is socially disenfranchised.

##### Not Making Others Feel Bad

Part of prioritising good feelings is to not make anyone else feel bad. Protecting his mother from worry was a key concern for teenager Wayin, and this informed his actions ‘I tried not to show what was happening to me. So, I tried to keep it hidden. Tried to get rid of it quickly’. A participant described supporting his young son with ARF by always giving positive feelings to mitigate bad feelings of stress, distress and worry about what could happen. In the context of community life generally, Yolŋu are always concerned to not cause offence, upset or in any way be the cause of bad feelings in relation to each other. Especially in a small community, taking care of others’ feelings is an essential means of ensuring community cohesion; the imperative to maintain the status quo is paramount. Conversely, good feelings are equally collectively experienced. 

#### 3.3.3. Specific Feelings 

This section describes three specific types of feelings evidenced in the analysis: two negative feelings and one positive. 

##### Shame and Shyness 

It is important to note that ‘shame’ in an Aboriginal context is a powerful emotion resulting from the loss of the extended self: experienced by, or for, a person who acts or who is forced to act in a manner that is not sanctioned by the group and that is in conflict with social and spiritual obligations. The fulfilment of obligations to the group is more important in Aboriginal society than isolated individual behaviour, especially individual assertiveness; group cohesion (is an) expression of life itself ([18], p. 598). Therefore, ‘to feel inferior and shamed is injury’ ([19], p. 182). 

Shame is often experienced as part of the lived experience of having ARF/RHD the result of ‘being singled out’. Wayin experienced significant physical effects (Sydenham’s chorea) as a result of the neurological effects of ARF. At the time of his interview, despite considerable improvement in his symptoms, Wayin described at length his enduring shame at still not being able to talk clearly. Shyness also made it hard for patients to tell clinicians how they prefer to have their injections, and shame affected participants going to the clinic. For instance, one grandmother described advising her friend whose child had ARF symptoms ‘I asked her to ask the doctor, he will explain it to you. She said ‘No, I am ashamed’.

##### Worry, Stress, Fear, Anxiety

There were many examples of participants describing a range of negative emotions encompassing the stress, distress, worry and trauma associated with experiences of ARF and RHD. Some are direct impacts, such as the worry about how to take care of a child with RHD; others are the result of adding RHD into already complex and stressful circumstances, such as crowded, inadequate housing (lacking hot or running water, and poorly maintained public housing) and poverty. These more general stresses converge with the fears associated with RHD. 

All carers expressed high levels of stress, grief and worry regarding finding out that their child had ARF/RHD. For many, the anxiety was associated with the many worries to do with children’s health and other family concerns; in the most extreme example, it was associated with the fear of the child being removed by child protection services. For some, it motivated them to find out all they could about the RHD, but uncertainty can create a constant state of vigilance; for instance, regarding his son’s ARF diagnosis, Lundu said ‘It give me worrying…all this sickness, we don’t know where they [the causes] are, therefore we need to look after our body all the time’. While Lundu had been given some information about the causes of ARF, it was communicated in a manner that produced fear. He described being given information that ‘made everyone scared’, he also described how he was motivated to take action because of the fear, indicating that from the health practitioner’s perspective, understanding relates to patients/parents/carers being motivated by fear.

Recommendations for caring for a child with RHD place an additional burden on families, particularly in the context of having many caring responsibilities. Some of this stress for carers may be amplified when supporting a teenager, given their need for independence and agency regarding their health. The stress of sickness may cause family members to lose connection, and sense of trust with each other, in turn impacting the well-being of all. 

##### Confidence 

Confidence was a frequently discussed positive emotion. Confidence is a key concept and is, therefore, intertwined with all three themes. That is, confidence is related to knowledge, understanding and the ability to make good choices, as a male co-researcher summarised, ‘if you have confidence you can follow your feelings, trust your feelings, and be able to share your feelings with family and friends’. 

Confidence is described in terms of being strong in mind and feeling; it is a strength that aids clear understanding. To think clearly prompts making good choices, as Lundu said ‘Yolŋu have to be strong, in our mind, in their feeling as well, and also in their reality, like in practical way of living’. Having confidence means one can assert one’s needs, such as asking for the antibiotic injection to be delivered in a preferred way.

##### Confidence Can Be Built 

Confidence can be built through actions, such as taking on challenges such as a new sport, using unfamiliar words/knowledge or even acting against medical advice. As Rinytjan stated ‘As I was growing up… I didn’t think about it too much, that I had a heart disease… I did sport and all that stuff… [it] helped that fear to know that I can do anything what other kids can do. So that made me more stronger, built up my confidence.

Additionally, good support, including role models, can make one stronger and more confident. Feeling better about oneself helps one make good choices. Therefore, building confidence was described by a co-researcher as ‘a way to guide teenagers rather than telling or forcing or making them scared’. 

##### Faith as a Source of Confidence and Inspiration 

Many Yolŋu have a strong Christian faith, and there were several examples where faith was a source of good feeling and confidence. For instance, Christian faith was used to explain a cure based on the understanding that God works through people. Lundu stated that ‘he [God] is working now today, with all of us, working through the people, to share the knowledge, to let individuals know about this particular [RHD] story I feel very confident, stronger through Him, through the one who is really doing the job for us, to fix the worrying’.

The positive feelings engendered by religious faith help to mitigate the fear of the unknown, especially in difficult times associated with sickness. In an uncertain, changeable world and without the confidence that comes from having a clear understanding, having faith mitigates worry, in turn enabling trust in doctors and medicine. However, while faith may help people feel better, it does not necessarily help to understand. Faith may at times be exploited by health practitioners, as simple faith and trust do not require knowing the full story. 

### 3.4. Creating Clear Understanding 

The theme ‘creating clear understanding’ includes two sub-themes: barriers to understanding and, conversely, what helps ‘good’ understanding. In terms of health care, this theme is important in suggesting that clear understanding relates to broadening the space of productive dialogue through positive experience and information coming from the right person with the right feeling. Further, it is important for some that the information is conceptually meaningful (the ‘deep inside story’) and sufficiently detailed. 

#### 3.4.1. Barriers to Understanding—‘I Don’t Know How We Could Stop the Sickness’

##### Signs

Being able to see and understand signs is fundamental to traditional experiential learning [20]. Knowing and understanding signs in the environment and interpreting them correctly to make good decisions is crucial to survival. This knowing is embedded in learning by experience, based on observations from watching others (leaders) follow signs and make choices. Information is provided as needed. In this context, there is little sense of uncertainty. There is also no need to ask questions, and there are even prohibitions against doing so in many instances. That is, one cannot ask for knowledge until one is given permission to do so.

In contrast to observable, interpretable signs (albeit ones that might take years to learn to recognise), biomedicine organises a complex repertoire of disease signs and symptoms into calculations of risk. Referencing relatively obscure technical knowledge, risk permeates biomedical thinking and practice. Demonstrating the profound difference in worldviews, the abstract noun ‘risk’ is not in the Yolŋu lexicon. 

Yolŋu wish to make meaning (as they would usually do easily) out of new signs and health situations: this is exemplified by the father of a child with RHD stating ‘We are the intelligent people’. Difficulty arises when new signs occur, such as those of RHD, which are unfamiliar and thus disconnected from actions. Not seeing the signs means not knowing, in turn, not knowing how it feels and not knowing the choices subsequently impact actions. In this context, framing health information in terms of ‘risk’ is of little conceptual use to Yolŋu. The difference in worldviews described here helps explain why clinicians get frustrated and why patients do not have confidence that they are getting good care. Not being able to interpret signs effectively, the Yolŋu are then faced with needing to learn how to acquire knowledge by asking questions and interpreting answers framed in the technical biomedical language of risk. Having to both develop new learning approaches and acquire new knowledge is uncomfortable, shaming and produces the kind of bad feelings that Yolŋu seek to avoid at all costs. Concurrently, it is not possible to separate the different patterns of disease and wellness from the realities and legacies of colonisation, where lifestyle has changed dramatically with the attendant social and economic disadvantages experienced by Yolŋu. 

##### What Is the Name?

A related issue was that the unknown names of diseases and germs were seen by Yolŋu participants and co-researchers as a specific piece of missing knowledge. The conceptual significance of ‘names’ is related to identity and place, as a male co-researcher stated ‘our names are in the songlines, and is also part of the land and the sea’. Aboriginal worldviews, ways of knowing, being and identities mean that the land is imbued with stories, songlines and knowledges [21,22]. Thus, the lack of ‘names’ and taxonomy is a significant conceptual gap, not just related to language and not solvable just with translations. Names provide an identity that makes something real and allow Yolŋu to begin to place and make meaning for germs, for example, within their worldview. The lack of names is worsened by the fact that germs are invisible to the naked eye and, therefore, have mysterious signs. 

##### Lack of Understanding about RHD and Its Signs

The qualitative data reflect that limited understanding of new sicknesses is more than just a language gap and, not surprisingly, that the Yolŋu generally consider sickness not to be a problem before *Balanda* arrived. More broadly, the Yolŋu do not have access to Western conceptualisations or theories of disease except, in some cases, to suggest that disease is the result of colonisation. This further suggests, therefore, that Yolŋu have not been given adequate explanations that match their worldview regarding causal connections of diseases that are new to them. When health information does not reflect Yolŋu ways of knowing, it leaves Yolŋu to try and make sense of and interpret their own signs, including *galka* (sorcery) in some cases. 

Within the context of a general unfamiliarity with new diseases, RHD is a particularly and intrinsically complex and confusing disease (see Table 1). In relation to RHD, there were many examples of language related to ‘signs’ and not being able ‘to see’, with participants speaking of ‘missing the sign’ (Dhumdhum); ‘confusing signs’ (Larrani’s mother); not seeing the signs (of death) (Dankapa); and not being able to see the ‘signs when we are sick’ (Mungudjurk). It is, therefore, difficult for Yolŋu to apply traditional methods of learning from experience (‘seeing the signs’) and sharing knowledge through storytelling regarding RHD. 

This is exacerbated by poor communication by clinicians and the tendency to focus on curative/acute care rather than prevention. Consequently, most Yolŋu began their ARF/RHD experience with limited and insufficient understanding, and some were not even sure if they or their child had had RHD or not. Other participants had complex narratives regarding the onset of their ARF, often conflating onset symptoms with other potentially related events (causes/signs). This confusion reduces the confidence to make decisions about treatments. Given traditional ways of knowing, contextual factors such as signs or reasons for sickness can seem more compelling than medical factors. 

##### Specific Confusion about Injections 

Consistent with other RHD research [23], confusion about the injections was common both in terms of their value or purpose and uncertainty about how long regular injections were required and when they might finish having them. Similarly, there was confusion about the impact of not having the injection. Questions about injections were often asked in interviews—reflecting that they had not had this information clearly communicated by clinicians. 

##### Signs Not Clear to Health Service Providers 

Moreover, it was evident based on the experiences of Yolŋu that some clinicians also may not recognise the signs of RHD. This is particularly poignantly demonstrated in the cases of three participants Wayin, Dhumdhum and a young woman, Yalku, with family histories of RHD, where, despite this information being available to clinicians, the patients experienced the devastating impacts of delayed diagnosis. Wayin described how he was set back two years because of the delayed diagnosis of his Sydenham’s chorea, stating ‘They could have listened to my mum straight away. Because they took out my appendix. but there was nothing wrong with it, my mum was telling the doctor [about RHD]. and one of them decided to listen but it was a bit too late. That’s when I lost my reading and that’. 

A key issue here is the lack of cultural and medical orientation regarding RHD for health service providers entering remote communities [24]. The failure to have a systematic approach to the induction to working in the NT can result in dire consequences.

##### Do Not Remember, Do Not Know, Forgetting 

As discussed in Theme 1, negative feelings, such as fear, anxiety, worry, stress and trauma, are to be avoided. While forgetting or not remembering may be the result of other factors (competing demands, low priority or trauma), it is also likely a means to avoid bad feelings. In turn, forgetting can cause confusion or the sense of lacking understanding. Similarly, as negative feelings contribute to confusion and feeling frightened, limiting the ability to engage with health professionals, particularly as ‘talking to doctors and nurses is something we Yolŋu find hard’ and, while an Indigenous Liaison Officer may be there to provide support, they are often busy. 

##### Reluctance to Answer Questions—‘Why Do Doctors Ask So Many Questions?’

As discussed in Theme 1, participants’ reluctance to feel bad by talking about being sick sometimes made it difficult to ask questions about health. In many medical situations, health service providers ask similar questions. A lack of understanding makes this reluctance worse; a co-researcher questioned ‘Why do people ask questions about the different kinds of sickness that Yolŋu don’t know? How can we when we don’t know the words?’. Not being sure of the words (names) causes a lack of confidence, so that even when there is some knowledge, people might not speak for fear of making a mistake and asking questions about a Western/biomedical condition in English results in negative feelings of shame, confusion and fear. 

#### 3.4.2. Factors That Help in Creating Good, Clear Understanding 

Evidence was provided about the kinds of knowledge and ways of knowing that contribute to clear understanding and, in turn, build confidence. These relate to building confidence and understanding through asking the right questions and experiences (including research). Information becomes culturally embedded when it came from the right person, with a good feeling (from the heart), was conceptually clear (the deep inside story) and was community related. Further, as soon as a conceptual understanding was developed, Yolŋu community members were keen to share the story of their new knowledge with family from other communities, reflecting both conceptual confidence and the manner in which knowledge was previously shared through storytelling.

##### Feeling Confident and Empowered Impacts on Ability to Ask Questions 

Traditional learning is observation and experience-based, and asking questions, unless given permission to do so, may result in being shamed for not having listened or observed well. Thus, asking uninvited questions as a means of acquiring knowledge is a relatively new, unfamiliar and culturally inappropriate method, consequently requiring considerable confidence. Guya insists that we ‘teach the little ones, make sure you ask questions… I tell him he has to speak, he can do it by himself’. For some, choosing to ask questions of health professionals helped build confidence, both through improved understanding and the practice of asking, which becomes a virtuous cycle—building further confidence. 

In some yarns, participants were confident to ask questions that they might not have asked in other circumstances, suggesting a safe space of productive dialogue gave participants a sense of being given permission to ask questions, resulting in some surprisingly basic inquiries, such as Mungudjurk’s father asking ‘When, how does it start, this sickness?’ Health practitioners may be unaware of the need to invite Aboriginal people to ask questions rather than assuming that they will, and then being critical when they do not.

##### Experience/Seeing/Knowing the Signs as Sources of Knowledge 

In contrast to the earlier discussion about the impact of not seeing or being able to interpret signs, some participants reported experiences that aided them in being able to do so. Participants Guya and Dhumdhum described how having experience helps to understand (and is an important source of knowledge). In fact, understanding signs as a result of experiences creates an imperative to share experiences and knowledge. Participants spoke of encouraging and supporting others through their stories. This, in turn, connects to Lundu’s frequent references to Yolŋu needing to learn together, such as ‘learn from him or from her, that’s how the confidence build up’. This demonstrates the Yolŋu capacity to communicate health information in a feel-good way.

##### Knowledge Needs to Be Family-Based and Community-Based 

It is often the case that the right person can help provide understanding in a way that enables another person to feel good. Given that it is usually family that plays the most significant role in creating good feelings, it is important that family members are supported to be confident in their understanding. Similarly, others recognised the importance of communicating health knowledge to communities rather than individuals, using community-level data [25,26].

##### The Role of Escorts 

The importance of having someone accompany Aboriginal patients to the hospital is recognised by health services. Escorts are almost always family members, demonstrating the importance of family in contributing to both feeling good and understanding. While their role is primarily to provide kinship (company), they may also play an important part in communicating information between health service staff and the patient. It is important that the escort is prepared and sufficiently confident to take on the role of asking questions; as Nyunyul asked regarding her grandson’s escort, ‘Do they know how to ask questions? Did they get the full story?’. 

Providing or conveying meaningful information while maintaining a good feeling is a challenging path to negotiate. The role of being an escort is fraught not only within the hospital but also when they return home, where there is often uncertainty about whom they can tell the patient’s health story to. The patient may not want to speak about their health problems, while other family members want to know and expect the escort to share what they know. Hospital confidentiality processes contribute to this confusion. This burden of responsibility is heightened if there is a possibility of the escorted person dying. 

### 3.5. Choosing a Good Djalkiri (Path) 

This theme brings together the previous themes of ‘maintaining good feelings’ and having a ‘clear understanding’ in terms of the decisions, choices and actions made by participants and their families. As described at the start of this section, the Yolŋu linked clear information (knowledge) and support to feeling good as being needed in order to make good decisions—to be on a good *djalkiri* (path) (Figure 1). From this straightforward beginning, the Yolŋu co-researchers analysed interviews looking at why people make the choices they do. Like many of the participants, the Yolŋu co-researchers were themselves grappling with how to make unfamiliar decisions. The nuances in the interconnections between the three themes became more complex as we dug deeper. Two sub-themes are explored here related to the question of who decides and using feelings to make decisions.

#### 3.5.1. Whose Choice Is It to Make? 

Yolŋu are not accustomed to making the kinds of choices necessary to manage complex, new health conditions. As discussed previously, traditionally, Yolŋu made choices based on correctly interpreting familiar signs and choosing between known variables, and these choices were, to a large extent, guided by feelings. That is, understanding the observed signs leads to confidence in making choices. This is contrasted with the language of risk used by health service providers, which leaves people feeling that there is ‘no choice’. Making choices can be empowering; the inability to make a choice in a medical context often results in the perception of being told what to do. 

##### ‘The Choice Is His’—The Individual Decides 

In a traditional cultural context, rather than instructing or ‘bossing’, families prefer to guide by gentle encouragement, and a strong theme in the interviews was that it is up to the individual to decide how to control his or her life. In order to illustrate how he would like his daughter to take care of her health, Gumbu’s father spoke of his self-determination regarding his own health care: ‘No doctor is going to tell me what to do… I have to make up my own mind’. Similarly, Rinytjan stated, ‘It’s something I give towards my children, I always step back, I don’t want to push them, I want to give them their own time’. While the expectation from health workers might be that encouragement in the context of unfamiliar health problems would be more overt, many traditional Aboriginal families are reluctant to intervene in a child’s development other than by showing pleasure in positive actions and growth.

##### Family Involvement in Making Decisions 

Conversely, families do have a degree of agency in decision-making for a sick person, both pragmatically (getting to appointments) and through the provision of good feelings—emotional support through encouragement and confidence-building. That is, for Yolŋu, the right person can help give understanding in a way that helps the person feel good, and family is more important than doctors in helping build confidence. Therefore, it should be expected that Yolŋu are more likely to seek help from family than health professionals. This principle extends to using family members as escorts to accompany patients to the hospital.

Family engagement will often involve negotiating the line between bossing and encouraging, with grandmothers tending to have more capacity to ‘boss’. In the case of Dankapa, there was much to-ing and fro-ing about the extent to which her heart surgery was her choice or the family’s—more specifically, her husband’s—decision. 

While there were examples of teenagers taking themselves to the clinic for their injection, for most young people family involvement was crucial in accessing care. Attending a specialist appointment from a remote community involves days of travel, meaning carers must take leave from work and leave siblings in the care of other family members. For some carers, this is too difficult, and other family members step up. 

##### Teenagers with RHD—The Challenges of Transitioning to Adulthood 

Regardless of the degree of family input into the care of a child, transitions in responsibility for decision-making take place as children become teenagers. For many Aboriginal children, this is likely to be at a younger age than non-Aboriginal children. Teenagers face the challenges of peer pressures, less parental guidance, greater involvement in conflicting cultural requirements of two worldviews, more exposure to ‘town’ and its temptations. These all impact health decision-making ([14] p. 178 for case studies).

Further, the transition from child to teenager is a crucial part of the RHD journey [23]. Of particular concern is that at 18 years old, teenagers leave the relatively specialised care of paediatric cardiology and move to the care of adult cardiologists who have less knowledge and experience of both RHD and young people. These young people then turn up in hospital in their mid-twenties, critically ill. The transition from paediatric to adult care is always difficult, but it is particularly so in the case of RHD as the period between the ages of 15 and 25 is a critical, high-risk time (Dr Remenyi, paediatric cardiologist, personal communication). 

##### Health Service Providers’ Role in Decision-Making 

The general reluctance to tell others what to do makes Yolŋu extremely sensitive to feeling they have been told what to do by *Balanda*. This was clearly stated by a co-researcher as ‘we don’t want to listen to that’. ‘Balanda can’t tell us what to do, can’t change our lives, we have to think clearly for ourselves’. This is particularly the case when what they hear from the clinician is ‘there’s no choice for you’. At the conclusion of this study, another co-researcher stated, ‘we do need (lifestyle) information but given in a way that makes us feel good. So, we can work together to solve problems, and help everyone to be healthy… then the choice is ours to make, to think about the decision’. 

Being disempowered in one’s decision-making by health service providers is particularly problematic for teenagers. 

#### 3.5.2. Using Feelings to Guide Decisions 

As discussed previously, RHD is a relatively new disease for Aboriginal people, creating uncertainty, lack of confidence and worry, consequently influencing choices and actions. Having to develop new learning skills and conceptual frames and acquire new knowledge makes it likely that some people will avoid making choices at all, particularly if they have already been made to feel bad regarding their health or experienced shame in interactions with health service providers. Alternatively, there are examples where more positive health-promoting actions resulted from being supported and encouraged, confirming that feelings are the most significant driver of decisions about actions. 

##### Encouragement and Support 

Encouragement was defined by a Yolŋu co-researcher as ‘giving support so he can make the right choice’. That is, feeling better about oneself can help one make good choices for oneself. Therefore, building confidence ‘is a way to guide teenagers rather than telling or forcing or making them scared’. The significance of the link between maintaining good feelings and health behaviour change was validated in a discussion about this study with a senior, very experienced Yolŋu researcher, Dr Lawurrpa Maypilama. 

Encouragement can also come from passing on knowledge based on one’s own experiences. In the course of the study, many of the teenage participants expressed interest in meeting other teenagers, describing how they had found support from friends or provided support to others. Further, some teenage participants, such as Wayin, Yalku and Miyapunu, also expressed the desire to support others more generally through sharing their stories as a community champion or mentor or, in Mungudjurk’s case, making a documentary about their RHD stories. 

##### Confidence 

As discussed earlier, confidence is a positive emotion related to a having clear understanding (appropriate action identifiable from a familiar sign) and, therefore, the ability to make good choices. Making choices from a place of feeling good is likely to set up a positive feedback loop. With confidence, trust in one’s own resilience, one’s capacity to deal with troubles is bolstered. Confidence builds as people make choices, take on challenges and notice their own strengths. With regard to making health decisions, Yolŋu and Western worldviews align regarding the confidence placed in those who have deep knowledge of signs (experts). That is, clinicians who clearly state their learning and depth of knowledge are likely to be considered worthy of trust, even if they lack the skills to communicate a clear conceptual understanding. 

## 4. Discussion

Strengths-based practice is not simply a ‘culturally acceptable’ way for non-Indigenous peoples to work for Indigenous peoples, but rather it is the only way of working with Indigenous people [27].

Adopting decolonising research methodologies and methods situates this research into a broader historical, political and cultural context [14,24,28]. Consistent with using a social constructionist theoretical approach to interpret the social and cultural experience of RHD disease, diagnosis and illness management among Yolŋu, the thematic analysis reported here privileges Aboriginal voices and reflects both rigorous Western academic qualitative, ethnographic methods [29] and Aboriginal ways of knowing, being and doing [30,31]. 

Offering new insights, the themes unpack ideas around the importance of feelings and how they relate to health-related choices and actions. The findings from this research are deeply embedded in a worldview that prioritises relationships and the wellbeing of family and community (maintaining a collective ‘good feeling’). Strong Aboriginal voices describing the significance of the inter-connectedness of knowledge, choice and behaviour/action have not previously been heard in the RHD domain, providing new social meaning for practitioners and policy-makers [29]. Further, our findings are likely to be applicable more broadly in other First Nations health contexts across Australia and in other colonial settings. The Yolŋu co-researchers were not in the academic space and did not have access to the phrase ‘Aboriginal ways of knowing, being and doing’ [30]. The similarity between this phrase and the themes described here further validates the generalisability of our findings.

Differences between the Yolŋu and Western/biomedical worldviews help explain why clinicians get frustrated and why patients lack confidence that they are getting good care or do not engage fully with treatment options. These differences can cause stress and fear, such that poorly delivered health messages can make a person feel bad (loss of confidence) to the point they might ask ‘are they *[Balanda]* trying to kill us?’, and result in a reluctance to talk about sickness or to ask questions. For the Yolŋu, good or bad feelings and clear understandings or confusion lead to good or poor choices or inaction, which become even more complex given the socioeconomic, political, historical and cultural contexts that impact their everyday lives. Worldview ‘differences’ remain unresolved as the ongoing colonisation of Yolŋu ways of knowing, being and doing reinforces negative stereotypes of First Nations people as being unable to make decisions about their health, in turn, justifying intervening in First Nations communities. The failure of the health system to acknowledge the ongoing colonisation inherent in a narrow biomedical knowledge and Western way of being and doing limits the space for productive dialogue [28]. 

From the 1920s, spanning a period of more than fifty years, Yolŋu children were taken from their homes, which is one of the darkest chapters in Australian history. In more recent years and since the NT emergency intervention, children have continued to be taken away, reinforcing the fears among families that the stolen generation continues. The legacy of this injustice continues to affect Yolŋu attitudes to the Balanda health care system and beyond. The fear expressed by Yolŋu in the interviews confirms the need for healthcare providers to understand and acknowledge this history and to work differently to establish trust and build relationships with Yolŋu. 

In other words, our results point to the need for healthcare providers to gain a deeper understanding of First Nations ways of knowing, being and doing, including acknowledging the legitimacy of knowledge and practices to promote healing. Creating the space for productive dialogue around the prevention, diagnosis and treatment of RHD requires approaches that focus on the strengths, capacities and capabilities of First Nations peoples, strengths inherent in ‘the structure and character of social relations, collective practices and identities’ ([32], p. 1405). For example, as identified in our study, maintaining a sense of wellbeing (good feeling) within family and community. Beyond strengths-based approaches in clinical settings, First Nations leadership and governance should be ‘mandated in health care planning, and be legitimised and given the authority as opposed to advisory roles or tokenism’ [33]. Recent policy reforms ( (accessed on 15 February)) strengthen First Nations decision-making leadership and governance by mandating data sovereignty and co-design principles.

### Recommendations

The Yolŋu co-researchers were clear that the purpose of their work was to ‘make a mat for everyone to sit on’. Being invited to *nhina* (sit) is the first step in learning together [1], demonstrating respect and relationship. Thus, Yolŋu were indicating that they were not only making practical suggestions about changes, but they also wished to do so in a respectful, non-confrontational way; that is, to encourage productive dialogue. They were motivated to share knowledge about what actions they wanted ‘with everyone, so everyone can benefit’.

During the thematic analysis, the Yolŋu described recommendations for clinical practice. Fundamentally, to encourage confidence in decision-making, health messaging needs to take into account the feelings of the hearer by being positive, strengths-based, not didactic or focused on deficits.

‘To hear and understand a clear story people need information but given in a way that makes us feel good, then the choice is ours to make, to think about the decision’ [34].

For example, given the distress caused by using the unfamiliar language of risk, clinicians could instead ask patients whether the information has been explained in a way that enables them to feel confident they understand what they need to do to stay well or not get sick. Further, in this context, trauma-informed care needs to be built around recognising the strength in choosing to feel good as an assertive act, particularly when one is socially disenfranchised. Finally, health information is likely to be more accessible if it focuses on the collective good of communities and families, and the language of ‘self-management’ is replaced with terms such as community care or community development.

As reflected in the findings, working with young people is an area for further exploration in the RHD context. This was identified as a particular gap in the reviewed RHD literature [10] and is at odds with the observation that teenagers are often the carers of younger children and babies who are most susceptible to harmful effects of Strep A infection. A greater focus on children and adolescents would build on the cultural strength that recognises the place of children in the continuity of First Nations societies as they grow up to be adults with the responsibility of caring for their culture and country [35].

## 5. Conclusions

Our research reveals the strong voice of Yolŋu co-researchers regarding improving health service delivery for Aboriginal children, families and communities. The depth and quality of the meaning-making and theoretical understandings that emerged from the collaborative research analysis and interpretation with the Yolŋu co-researchers highlight the importance and value of productive dialogue in the intercultural space.

The application of the findings and conclusions of this paper have important implications for further research with First Nation’s communities in other countries with a similar colonial history. The decolonising, constructionist research approach adopted in this study [28] confirms that illnesses such as RHD, which remain a significant burden for disadvantaged and marginalised groups globally, are particularly embedded with cultural meaning that, in turn, is impacted by biomedical understandings and responses that differ greatly from those with lived experience of that illness. Ensuring that the experiences of children, adults and communities with RHD become meaningful tools for advocacy and action requires critical reflection about power differences by health practitioners [36] and resourcing and co-designing health systems to prioritise local community input consistent with recent policy reforms outlined in the Close the Gap National Partnership Agreement [37]. This shift in focus is likely to highlight a tension between Western, rationalist theories of behaviour change that do not align with a worldview that prioritises maintaining the good feeling and well-being of all. However, any discomfort caused by this tension should be valued as it is only when the productive potential of difference is emphasised that the actual work of collaboration is achieved [1,38]. That is, discomfort is a positive indicator of both power shifts and full engagement in a both-way learning, productive dialogue process that results in understanding our differences without diminishing either identity or empathy.

## Figures and Tables

**Figure 1 ijerph-19-04650-f001:**
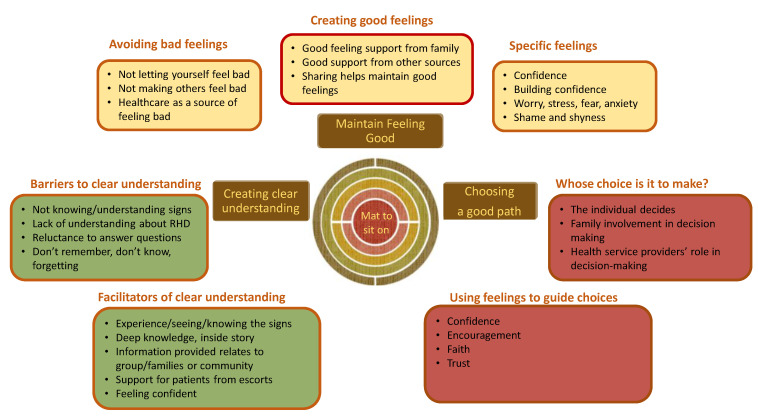
Thematic analysis summary: a mat for everyone to sit on.

**Figure 2 ijerph-19-04650-f002:**
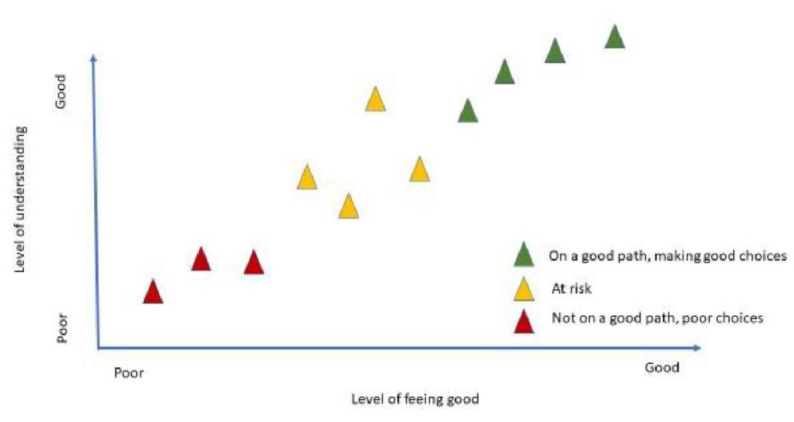
Good feelings based on clear understanding led to good choices.

**Table 1 ijerph-19-04650-t001:** ARF/RHD Disease Progression and Prevention Strategies/Treatment.

Disease Stage	Average Age Affected	Prevention Strategies/Treatment
1. Exposure to Group A streptococcus causes sore throats and skin sores	From birth	Reduction in household crowding, poverty and malnutrition Improved access to health care Prompt treatment with antibiotics to control infection
2. Acute Rheumatic Fever (ARF) Recurrences further damage heart valves	Initial episode most common in 5 to 14-year-olds	Treatment with antibiotics—normally monthly penicillin injections for 10 years
3. Rheumatic Heart disease (RHD) Chronic heart valve damage	Can begin in childhood, increases with age, incidence peaks between 25 and 40 years	Continued regular antibiotics for people at risk of ARF recurrence Regular specialist appointments to monitor heart
4. Heart failure (complication of RHD)	30% of those with RHD progress to heart failure within 5 years of diagnosis	Medical management of symptomatic RHD Open-heart surgery to repair/replace valves

(Adapted from [2,8]).

## Data Availability

Thesis contains more detailed data, full de-identified interviews available on request.

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
