# Peer review of "Living with Rheumatic Heart Disease at the Intersection of Biomedical and Aboriginal Worldviews"

_ijerph, 2022, doi:10.3390/ijerph19084650_

Round 1
Reviewer 1 Report
General comments:
In their manuscript: “Living with Rheumatic Heart Disease at the Intersection of Biomedical and Aboriginal Worldviews” Haynes E. et al. used qualitative interview data from 24 First Nations Australians to investigate health communication strategies in the Rheumatic Heart Disease (RHD) domain. The study identified fundamental differences between Aboriginal and biomedical worldviews contributing to the failure of current approaches to communicating health messages around RHD and provides relevant recommendations for culturally-responsive health encounters.
The manuscript offers findings to gain a deeper understanding of First Nations ways of knowing, being and doing in the context of healthcare and health service delivery, wherefore I would like to suggest the study for publication.
For the discussion I suggest to use and implement social constructionist theory to interpret the social experience of disease, diagnosis, and illness (see for example: J Health Soc Behav. 2010;51 Suppl:S67-79).
Author Response
Dear reviewer, thank you for your encouraging and constructive comments.
We have provided a point-by-point response to all reviewer’s comments in the attached table.
kind regards,
Emma Haynes on behalf of co-authors

Reviewer 2 Report
The paper starts out from the premise that Indigenous Australians (Yolŋu) are among the groups most severely affected by Rheumatic Heart Disease (RHD). However, Yolŋu have not been able to benefit from adequate health care solutions. The paper is based on an empirical study aimed at finding out why this is the case. On the basis of interviews conducted with both RHD patients and their caregivers, it systematically charts a disconnect between Yolŋu world views and the epistemologies of Western biomedicine and medical care. It then goes on to describe measures that might be taken so that Yolŋu ways of “knowing, being and doing” may be integrated into health care. Through such an intercultural form of medicine, health care systems would be reformed in a way that Yolŋu would able to benefit from treatment options and would be able to help prevent RHD in their communities. This would lead to a better, more culturally sensitive health care system in Australia and beyond.
Comment: This paper is tremendously important. On the basis of a single empirical study, it charts a terrain that is highly relevant not only for indigenous communities in Australia, but also far beyond; it can be applied to indigenous communities in other settler colonies such as the US, Canada and New Zealand; it is also of significant value for intercultural medicine in general. It argues that ultimately, health care will not be effective unless it takes into account the epistemologies and world views of different patient communities, both First Nations communities and, it could be argued, also migrant communities. One of the main strengths of the paper is its systematic description of Yolŋu world views, which it translates and describes for non-Yolŋu readers. The paper employs a decolonial methodology both with regard to the empirical study it is based on, and the argument it weaves. It creates a dialogue among Yolŋu voices, drawing the reader into the Yolŋu community. It makes a compelling case for why Western health care system have not been able to reach Yolŋu communities, since they are based on different epistemologies with regard to the questions asked by clinicians, with regard to an individualist rather than communal view towards sickness and care, and in its evoking of risk.
Suggestions for revision: The following suggestions are meant as possibilities for enhancing the coherence and readability of the article; its premise is both well-taken and important, so these suggestions only concern the “fleshing out” of the argument itself. Two aspects seem especially important in this context:
- The article’s authors might want to indicate more clearly where quotations start. Quotations from secondary sources as well as from interviews could be more visibly separated from the main text, so they would be more clearly visible as quotations. These quotations could also be introduced in a more explicit way, e.g. “As health care provider x suggests…”, “As co-researcher x states”… At present, the article weaves these voices together, which, on the level of argument, is highly fruitful; however, it may detract from the readability and accessibility of the article.
- One of the main strengths of the article is the way it draws on the qualitative interviews conducted with patients and caregivers from the Yolŋu community. However, the interviewees might be more directly introduced, e.g. “This is also addressed by Dhumdhum: (quote).” Moreover, patients are referred to both by their names and sometimes by their role in the family, e.g. Dhumdhum’s father or (name). This can be confusing to the reader. It would also enhance the argument if a little more detail could be provided on the people who are being quoted, such as x, a fifteen-year old teenager who has come to the interview with his aunt, … So rather than saying, “…or even refusing to answer questions, such as when Dankapa (patient)…,” the patient might be put at the center of the sentence, e.g. “Patients might also simply refuse to answer questions. For instance, Dankapa suggested in an interview…”
- Especially the first chart is highly valuable and may serve as a guide through the entire argument; it is also highly valuable in that it shows the extent to which the different views and concepts are interconnected. The second graph, however, does not seem as easily accessible; the authors might want to consider omitting it from the article, or else they may want to describe the reasons for its inclusion. Unlike the first chart, it is not self-explanatory by simply looking at it.
Minor suggestions:
- The role of decolonial methodologies might be briefly described and a reference could be provided, such as Linda Tuhiwai Smith’s work, but also First Nations theorists from Australia.
- One of the strongest points made in the article is that in Yolŋu communities, there is often a fear of clinicians since historically, clinicians were also present when Yolŋu children were taken away from their families. For readers not familiar with Australian history, one or two sentences might be added here to shed light on the history of taking Yolŋu children from their homes, which is one of the darkest chapters in Australian history. It would then become clearer that the legacy of this injustice continues to affect Yolŋu attitudes to the Baldana health care system and beyond. In the US, for instance, the Tuskeegee experiments are partly responsible for the fact that African Americans mistrust the health care system and have been wary of getting vaccinated against the Covid virus.
The paper is highly important to the field of health care, intercultural health care, and indigenous studies. It makes many strong points that will be valuable for research and scholarship in all these areas. One of the questions that the authors might think about is whether the refer to the implications which their paper might have for further research, e.g. the applications of their findings to First Nations communities in other settler colonies.
Author Response

(The authors gave the same response as above.)

Reviewer 3 Report
Dear authors,
Thank you for the opportunity to review your paper.
It is of the highest calibre: interesting, important, well-written and with appropriate and novel methods.
Some very brief thoughts/ compliments:
- Your writing style is professional and fluent.
- Your portrayal of the topic and client group is sensitive.
- The topic regarding health care access and comfort and literacy of disease is important and needy of exploration.
- Your Methods section is great. I would suggest that yarns are not equivalent to interviews (albeit very similar) and that 'nhina, nhäma ga ŋäma' might not be equivalen to participant observation. It would be good to add some extra words to explain how they differ and why standard interviews and PO were not quite appropriate.
- The author team is comprehensive with so many aspects covered in their collective experience- culture, language, medicine, research, health systems understanding...
- The findings are highly relevant to this population and disease. Would you consider adding a little more about the relevance to other Australian Countries (beyond Yolngu), to other non-Inidgenous RHD patients, and to other country settings?
Overall, well done. A beautiful submission and worthy of publication.
Author Response

(The authors gave the same response as above.)
